# TRIDENT: The Nonlinear Trilogy for Implicit Neural Representations

## Abstract

Implicit neural representations (INRs) have garnered significant interest recently for their ability to model complex, high-dimensional data without explicit parameterisation. In this work, we introduce `TRIDENT`, a novel function for implicit neural representations characterised by a trilogy of nonlinearities. Firstly, it is designed to represent high-order features through order compactness. Secondly, `TRIDENT` efficiently captures frequency information, a feature called frequency compactness. Thirdly, it has the capability to represent signals or images such that most of its energy is concentrated in a limited spatial region, denoting spatial compactness. We demonstrated through extensive experiments on various inverse problems that our proposed function outperforms existing implicit neural representation functions.

## 1 Introduction

Implicit Neural Representations (INRs) signify a fundamental shift from conventional grid-based data structures. INRs represent a significant departure from the traditional, grid-based approach to data representation. Instead of storing data in fixed, discrete units such as pixels or voxels, INRs employ a neural network to encode data as a continuous function, symbolically represented as $f : \mathbb{R}^n \to \mathbb{R}^m$, where $n$ signifies the input dimensions (e.g., spatial coordinates) and $m$ the output dimensions (e.g., colour intensity in images). This enables INRs to capture complex details without being confined by grid resolution, making them particularly useful for various inverse problems, allowing for high-fidelity and resolution-independent processing. In contrast, traditional grid-based approaches rely on fixed, discrete grids—such as pixels in images or voxels in 3D models—leading to limitations in resolution and increased memory consumption for high-resolution datasets or complex shapes. Thus, INRs provide a more flexible, memory-efficient alternative, facilitating enhanced control and manipulation of data, applicable across a broad spectrum of fields.

Implicit Neural Representations (INRs) have rapidly gained prominence in diverse domains, such as medical imaging (Zou et al., 2023), computer graphics (Mildenhall et al., 2021), and robotics (Khargonkar et al., 2023), due to their remarkable practical applicability. INRs offer a unique approach to encoding complex data, leveraging deep neural networks to represent information implicitly and continuously. Researchers have undertaken substantial efforts to improve INRs, introducing various techniques from sinusoidal functions like SIREN to wavelet transforms like WIRE (Saragadam et al., 2023). Despite these advancements, existing INR techniques grapple with critical challenges.

Firstly, a fundamental issue arises from the specialised nature of existing INR techniques, which are often designed for specific tasks, resulting in a lack of adaptability and generalisability across diverse problem domains(Song et al., 2023). This inherent task-specific imposes limitations on their broader applicability, constraining their efficacy in addressing a wide spectrum of inverse problems.

Secondly, current INR functions have inherent difficulties when attempting to capture intricate fine-grained details while simultaneously avoiding the introduction of undesirable artifacts. Notably, they often encounter challenges in representing high-order information within low-order domains, which leads to a degradation of information fidelity and the generation of inaccurate representations. Furthermore, these methods may exhibit suboptimal performance in efficiently encoding frequency information and spatial regions, thereby impeding their effectiveness in representing complex data structures in several tasks.

In this study, we introduce a novel Implicit Neural Representation function named `TRIDENT`. Our approach is distinguished by a trilogy of nonlinearities, which set it apart from existing techniques in the field. Firstly, we incorporate order compactness into `TRIDENT`, enabling the transformation of high-order information into a low-order representation. This part enhances our function's capacity to represent complex data efficiently. Secondly, frequency compactness is another component of our trilogy, boosting `TRIDENT` to adeptly capture and represent frequency information, ultimately improving its fidelity in encoding intricate data structures. Lastly, our function offers spatial compactness, ensuring that it represents signals or images with a focus on concentrating the majority of their energy within limited spatial regions. This feature enhances `TRIDENT`'s ability to faithfully capture essential spatial features. Our contributions are summarised next.

☛ We introduce a novel function called `TRIDENT` for implicit neural representation, that stands out for:

- `TRIDENT` proposed a unique trilogy of nonlinearities being the first INR function that carefully is designed to enforce better approximations that generalise well across several inverse problems.

- `TRIDENT` proposed trilogy enforces: higher-order features (order compactness), better representation of frequency information (frequency compactness), and concentration on useful spatial regions for fine-grained details (spatial compactness).

☛ We demonstrate the capabilities of our `TRIDENT` function through extensive experiments across various inverse problems, showing that it not only outperforms existing methods in the literature but also generalises well across tasks, consistently delivering the best performance

## 2 Related Work

This section revisits existing literature on Implicit Neural Representations and the core principles of our novel technique, providing a basis for contrasting it with current solutions.

**Implicit Neural Representation.** An Implicit Neural Representations (INR) employs a neural network to model a continuous function that can interpolate or approximate the underlying data distribution of a dataset. In contrast to conventional grid-based approaches which discretely represent data, INR , built upon multilayer perceptron (MLP) networks, offer a continuous and differentiable framework for signal recovery (Yüce et al., 2022). It inherently differ from traditional Deep Neural Networks (DNNs) by focusing on learning from a single data instance, rather than relying on extensive training datasets. Previous research has demonstrated the versatility of INR in addressing a wide spectrum of challenges across diverse domains, ranging from medical imaging (Zou et al., 2023), signal processing (Sitzmann et al., 2020), occupancy volume (Mescheder et al., 2019) to computer graphics (Mildenhall et al., 2021).

The community has explored different alternatives to improve INR. The traditionally popular ReLU activation function fails to represent features in the high-frequency domain well (Hao et al., 2022), leading to low accuracy of ReLU-INR. Several improvements in network architectures have been proposed in recent years. Borrowing the idea from position encoding (Müller et al., 2022; Mildenhall et al., 2021), the authors of Tancik et al. (2020) proposed Fourier Encoding to represent high-frequency features in low-dimensional domains. Despite a significant improvement in representativeness, training with the ReLU activation function still results in low accuracy.

Inspired by the principles of Fourier Transformation, the authors of Sitzmann et al. (2020) introduced a network leveraging sinusoidal functions, referred to as SIREN, with the aim of enhancing the representational capacity of the network. Additionally, it is noteworthy that various other alternatives to nonlinearity have been under consideration including Ramasinghe & Lucey (2022). The Multiplicative Filter Networks, use sinusoidal functions and the Gabor wavelet to function as a filter to replace the activation function in INR architecture (Fathony et al., 2020) Analogous to SIREN, the WIRE model, as proposed in Saragadam et al. (2023), also adopts the Gabor wavelet as the activation function for the Implicit Neural Representation (INR).

**Inverse problems.** Inverse problems manifest in several crucial practical applications, spinning from image restoration to acoustic source reconstruction. The community has discussed rigorous mathematical

foundations (Devaney, 2012; Bertero et al., 2021; Vogel, 2002) for understanding inverse problems on imaging, from its ill-posedness (Rudin et al., 1992; Candès et al., 2006; Venkatakrishnan et al., 2013) to the parameter selection (Wei et al., 2022).

The classical approaches for addressing inverse problems are knowledge-driven. One category involves providing approximating solutions with boundary conditions that guarantee the existence of the ill-posed problems (Engl et al., 1996; Kirsch et al., 2011), while the other integrates knowledge of targeted parameter structures with sparsity assumptions (Daubechies et al., 2004; Jin & Maass, 2012) or stochastic models (Kaipio & Somersalo, 2007; Mueller & Siltanen, 2012). However, the classical methods have their inherent limitations. The former approaches suffers from limited physical priors from the observation and high computational cost for accurate analytical models, the latter is limited by its instability in capturing the task-specific data structures (Arridge et al., 2019).

Recent advancement in deep learning techniques also offers solutions on inverse problems with data-driven models (Ongie et al., 2020; McCann et al., 2017). They take data examples on specific problems with the prior knowledge which generates the data for training to learn the data structure, while in many applications the lack of sufficient data prevents it from being robust, thus insufficient to support an entirely data-driven approach. A recent avenue of progress is now combining the knowledge- and data-driven models to solve inverse problems. Several algorithms (Venkatakrishnan et al., 2013; Lucas et al., 2018) have been proposed for this combination, yet these findings predominantly remain experimental and lack a comprehensive theoretical underpinning. Nevertheless, there is an emergence of certain mathematical concepts for addressing data-driven approaches in the context of inverse problems (Arridge et al., 2019). The majority of data-driven techniques rely on a large amount of data, hindering their wide applicability. Implicit Neural Representations (INR) allow solving and deriving inverse problems from a single sample.

## 3 Methodology

This section details two key aspects of our `TRIDENT` approach: firstly, the formulation, and secondly, its theoretical foundation.

### 3.1 TRIDENT Formulation

In this work, we introduce a novel implicit neural representation function, which stands out for its several advantages over existing methodologies in the field. Our new function reads:

$$\exp(-|s_0([\mathbf{x}, \cdots \cos(2\pi\sigma^{j/m}\mathbf{x}), \sin(2\pi\sigma^{j/m}\mathbf{x}) \cdots]^{\mathrm{T}})|^2), \tag{1}$$

where $\sigma$ represents the frequency parameter, $m$ is the mapping size and $j$ stands for the mapping index, $j \in \{1, 2, ..., m\}$, while $s_0$ denotes the scale parameter. Our function equation 1 is then considered within our defined network architecture, which is structured as follows:

$$\Phi(\mathbf{x}) = \mathbf{W}_n \left(\phi_{n-1} \circ \phi_{n-2} \circ \ldots \circ \phi_1 \circ \gamma\right)(\mathbf{x}) + \mathbf{b}_n, \tag{2}$$

where $\sigma, W, b$ represent the activation function, weights, and biases of the network, respectively. Moreover, each element of the input vector $\mathbf{x}_i$ is transformed by $\phi_i$ as:

$$\mathbf{x}_i \mapsto \phi_i(\mathbf{x}_i) = \exp\left(-s_0|\mathbf{W}_i\mathbf{x}_i + \mathbf{b}_i|^2\right).$$

Moreover, the first-layer mapping, $\gamma$, is defined as: $\gamma(\mathbf{v}) = \left[\mathbf{v}, \ldots, \cos\left(2\pi\sigma^{j/m}\mathbf{v}\right), \sin\left(2\pi\sigma^{j/m}\mathbf{v}\right), \ldots\right]^{\mathrm{T}}$, and $\phi_i : \mathbb{R}^{M_i} \mapsto \mathbb{R}^{N_i}$ is the $i^{th}$ layer of the network.

### 3.2 Theoretical Foundation of TRIDENT

*What properties characterise our proposed implicit neural representation function, and how do they set it apart from existing models?* Unique in its approach, our function is distinguished by a set of theoretically-motivated properties. These properties not only endow our function with novel capabilities but also align it with

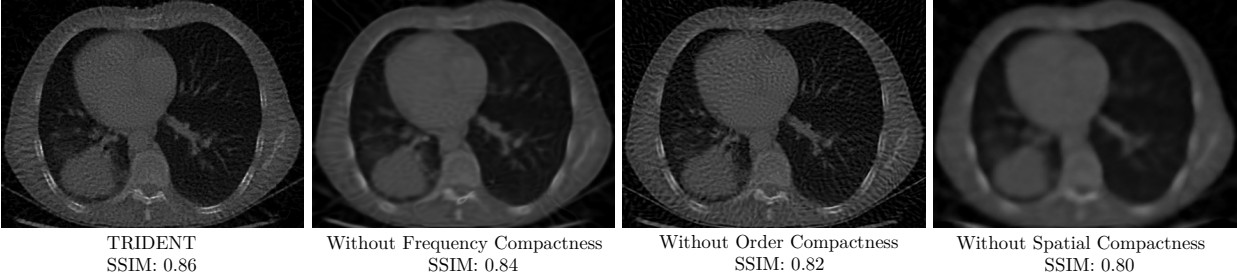

|  |  |  |  |
|---|---|---|---|
| TRIDENT | Without Frequency Compactness | Without Order Compactness | Without Spatial Compactness |
| SSIM: 0.86 | SSIM: 0.84 | SSIM: 0.82 | SSIM: 0.80 |

Figure 1: Visualisation comparison of CT reconstruction task on 100 projections among TRIDENT, and TRIDENT without Frequency, Order, and Spatial Compactness separately.

fundamental theoretical principles that have not been fully explored in prior work. Without losing generality, we simplify equation 1 to $\exp\left(-|\cos(x)|^2\right)$ and let $\phi(x) = \phi_1 \circ \gamma(x)$. We next introduce TRIDENT foundation based on a trilogy of nonlinearities.

↻ **Order Compactness.** The first concept we used is order compactness. It refers to the ability of a representation to encapsulate information efficiently across different orders of derivative or variation within the data. Moreover, order relates to the smoothness and the level of detail that the representation can capture. Mathematically, this can be associated with the representation's capacity to approximate functions and their derivatives to a certain degree of accuracy. In real context, a continuous function on closed intervals can be expanded as a series of cosine functions, e.g., $\sum_{n=0}^{\infty} A_n \cos \frac{n\pi x}{l}$. Meanwhile, known from the Stone–Weierstrass theorem, continuous functions on closed intervals can be uniformly approximated by polynomial series. Inspired by the above two theorems, we aim to enable our function, $\phi(x)$ to represent high-order information in the low-order domain, augmenting the representational capacity of our function. By elevating the series index $n$ to the exponent of the cosine function in the Fourier cosine series, we obtain:

$$\phi(x) = \sum_{n=0}^{\infty} A_n \cos^n\left(\frac{\alpha\pi x}{l}\right), \tag{3}$$

where $\alpha$ is the frequency coefficient and $l$ is the length of the domain. Without loss of generality, let $\alpha = \frac{2}{\pi}$ and $l = 1$, yielding: $\phi(x) = \sum_{n=0}^{\infty} A_n \cos^n(2x)$.

Consequently, with $\phi(x)$ as our formula, the network is enabled to represent high-order features, which we call *Order Compactness*.

↻ **Frequency Compactness.** In our work, we use the concept of frequency compactness to describe the efficiency with which a representation can encapsulate information at different frequencies. A representation with high frequency compactness can accurately reproduce both the overarching structures and the intricate details of the data. If a continuous function is expanded as a series of cosine functions, each appropriately scaled and shifted, it implies that this Fourier series provides an efficient way to represent frequency features, a concept we can refer to as frequency compactness. Consequently, if our formula takes the form as in equation 3, it will inherently preserve the strong frequency compactness characteristic of the Fourier cosine series, as demonstrated by:

$$\cos^n \theta = \frac{1}{2^n}\left(\begin{array}{c} n \\ \frac{n}{2} \end{array}\right) + \frac{2}{2^n}\sum_{k=0}^{\frac{n}{2}-1}\left(\begin{array}{c} n \\ k \end{array}\right)\cos[(n-2k)\theta] \tag{4}$$

which means that our formula, $\phi(x)$, like the property of the Fourier series, has the capacity to represent frequency information efficiently, bringing our method frequency compactness.

↻ **Spatial Compactness.** Another relevant concept that we use in our work is spatial compactness, which express variations within a specific, usually limited, spatial region. It is crucial in the context of INR as it determines how well the representation can be localise and render details within distinct pats of the space being modelled. Apart from the frequency compactness and the spatial compactness already brought by $\phi(x)$ in the form of equation 3, we aim to utilise the choice of $A_n$ to further enhance spatial compactness. This refers to the ability to represent a signal or an image in a way that concentrates most of its energy in a limited

spatial region. If we define $\alpha_n$ as $\alpha_n = \frac{(-1)^{n+j}}{2^{n+j} \cdot (n!) \cdot (j!)}$. Given that $|\alpha_n| \leq \frac{1}{2^{n+j}}$, and consider the convergence of $\sum_{j=0}^{\infty} \frac{1}{2^{n+j}}$ to $\frac{1}{2^n}$, the series $\sum_{j=0}^{\infty} \frac{(-1)^{n+j}}{2^{n+j} \cdot (n!) \cdot (j!)}$ is determined to converge uniformly by the Weierstrass M-test. Consequently, this allows us to express $A_n$ as:

$$A_n = \sum_{j=0}^{\infty} \frac{(-1)^{n+j}}{2^{n+j} \cdot (n!) \cdot (j!)}.$$

We then get:

$$
\begin{aligned}
\phi(x) &= \sum_{n=0}^{\infty} \left( \sum_{j=0}^{\infty} \frac{(-1)^{n+j}}{2^{n+j} \cdot (n!) \cdot (j!)} \right) \cos^n(2x) \\
&= \sum_{n=0}^{\infty} \left( \sum_{j=0}^{\infty} \frac{(-1)^{n+j}}{2^{n+j}(n+j)!} \frac{(n+j)!}{n!j!} \cos^n(2x) \right) \\
&\quad \text{Let } i = n+j, \text{then} \\
&= \sum_{i=0}^{\infty} \left( \sum_{j=0}^{i} \frac{i!}{(i-j)!j!} \cos^j(2x) \right) \cdot \frac{(-1)^i}{2^i i!} \\
&= \sum_{i=0}^{\infty} \frac{(-1)^i}{2^i i!} \left[ \cos^i(2x) + \binom{i}{1} \cos^{i-1}(2x) + \cdots \right] \\
&= \sum_{i=0}^{\infty} \frac{(-1)^i}{2^i i!} (1 + \cos(2x))^i \\
&= \sum_{i=0}^{\infty} \frac{(-1)^i}{i!} \left( \frac{1 + \cos(2x)}{2} \right)^i \\
&= \sum_{i=0}^{\infty} \frac{(-1)^i \cos^{2i}(x)}{i!}
\end{aligned}
\tag{5}
$$

Linking back to the exponential series: $\exp x = \sum_{i=0}^{\infty} \frac{x^i}{i!}$, we finally get:

$$\phi(x) = \sum_{n=0}^{\infty} A_n \cos^n(2x) = \exp\left(-|\cos(x)|^2\right) \tag{6}$$

which becomes our simplified formula. Thanks to the Gaussian window, we can represent the signal or image in a well-localised spatial domain. Consequently, our simplified formula enjoys the benefit of the nonlinear Trilogy: order, frequency, and spatial compactness.

However, it is well known that while the full Fourier cosine series effectively represents the even part of a function, it falls short of capturing the odd part. Our approach enhances the representativeness of odd components by incorporating dedicated sine channels. Additionally, to ensure the preservation of non-Fourier components, an input channel is integrated into our system, with which comes the three-channel formula: $\exp(-|s_0([x, \cos(2\pi\mathbf{x}), \sin(2\pi\mathbf{x})]^{\mathrm{T}})|^2)$. Finally, for a more efficient representation, we introduce a frequency parameter $\sigma$ and apply a log-linear strategy. This strategy is designed to balance magnitudes of different scales and aims to improve the output. Our function then reads:

$$\exp(-|s_0([x, \cdots \cos\left(2\pi\sigma^{j/m}\mathbf{x}\right), \sin\left(2\pi\sigma^{j/m}\mathbf{x}\right) \cdots]^{\mathrm{T}})|^2)$$

While Fourier series can indeed approximate arbitrary functions, the compactness properties enhance the efficiency and effectiveness of learning by reducing the model's complexity and improving generalisation. Essentially, our properties guide the selection of functions that the model can approximate well, focusing on those that are significant for the application.

We demonstrate the nonlinearity trilogy of our formula, which encompasses order, frequency, and spatial compactness. To visually illustrate the impact of each compactness type, we refer to Figure 1. The outputs in

Table 1: Properties comparison of the state-of-the-art Implicit Neural Representations against our proposed TRIDENT.

| Method | Principle | Disadvantages |
|---|---|---|
| SIREN | frequency compactness | instability |
| FourierNet | represent high-frequency features | frequency limitation |
| WIRE | wavelet transform | not good in high dimension |
| TRIDENT | order, frequency and spatial compactness | — |

Table 2: Table showing experiments' results of SIREN, WIRE, the baseline(RELU+POS.ENC.), MFN, and TRIDENT.

| METHOD | Denoising (PSNR) | | Occupancy (IoU) | Super resolution (PSNR) | | | | Audio (Loss) | CT Recon (SSIM) |
|---|---|---|---|---|---|---|---|---|---|
| | Parrot | Butterfly | Thai-Statue | Zebra | Baby | Monarch | Pepper | Friends | 100 projections |
| SIREN | 27.40 | 23.02 | 97.69 % | 23.04 | 28.99 | 24.21 | 26.63 | 0.000235 | 0.84 |
| WIRE | 30.05 | 24.65 | 98.79 % | 23.07 | 30.35 | 25.18 | 27.21 | 0.011515 | 0.82 |
| RELU+ POS.ENC. | 26.13 | 22.61 | 98.31% | 20.55 | 27.36 | 23.84 | 24.83 | 0.002822 | 0.80 |
| MFN | 30.20 | 24.83 | 97.15% | 17.21 | 21.67 | 21.51 | 20.39 | NaN | 0.68 |
| TRIDENT | **31.26** | **25.27** | **99.29%** | **23.26** | **30.56** | **25.34** | **27.27** | **0.000157** | **0.86** |

this figure were generated using the setting described in Section 4.6, using 100 projections. TRIDENT achieved an Structural Similarity (SSIM (Wang et al., 2004)) of 0.86. Removing the sinusoid channels reduced the SSIM to 0.84, impacting sensitivity to high-frequency features and detail preservation. We demonstrated the loss of high-order information without order compactness. Substituting the Gaussian window with ReLU activation, to test the absence of spatial compactness, resulted in an over-smoothed output with an SSIM of 0.80. These results underscore TRIDENT's advantage in representing both odd and even features efficiently.

## 4 Experimental Results

This section details all experiments conducted to validate our proposed TRIDENT function.

### 4.1 Implementation Details & Evaluation Protocol

For fair comparison in all experiments, we used a share code-base. Every experiment was conducted using the PyTorch platform and the Adam optimiser (Kingma & Ba, 2014). In this work, we consider five tasks: image denoising, 3D occupancy reconstruction, image super-resolution, audio reconstruction, and CT reconstruction. We evaluate each task using its corresponding metric, as stated in the respective task descriptions. Moreover, we adhere to a standard protocol by employing a grid search technique to identify the optimal hyperparameter values for our approach We therefore provide a set of recommended hyperparameter values that we found to yield the best results in each task.

If not otherwise mentioned, we use 2 hidden layers with 256 hidden features in each layer. Apart from the Audio task, we added the scheduler to all the other tasks to reduce the learning rate during training. We compared our function against the current literature including Positional Encoding (Mildenhall et al., 2021), MFN (Fathony et al., 2020), SIREN (Sitzmann et al., 2020) and WIRE (Saragadam et al., 2023). For SIREN nonlinearity, we set the frequency parameter as 10 as suggested in Sitzmann et al. (2020), and use the recommended parameters for WIRE from Saragadam et al. (2023).

### 4.2 Image Denoising

For evaluating our method's robustness, we reconstruct a colourful (with high-frequency features) image from the DIV2K dataset (Agustsson & Timofte, 2017). For orthogonality to the Gaussian window, we chose to add independently poison-distributed noise to each pixel, with a maximum average photon noise of 30 and an integration time of 2.

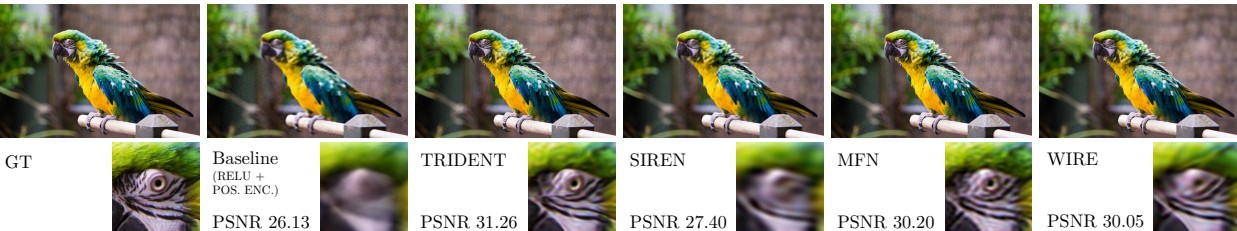

Figure 2: The visualization comparison of the denoising task on the "Parrot" example among the Baseline (ReLU + Position Encoding), `TRIDENT`, SIREN, MFN, and WIRE methods.

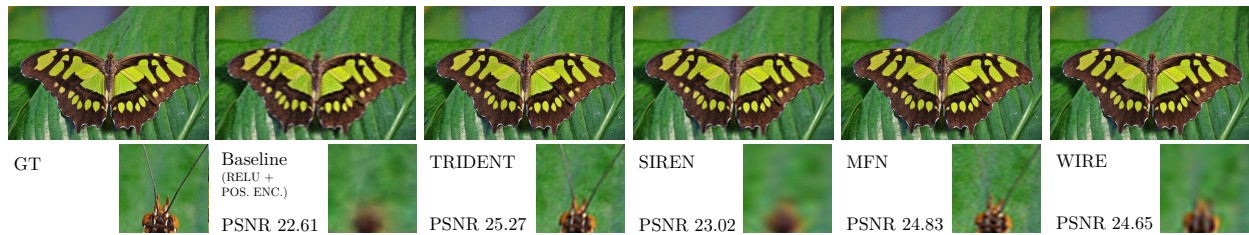

Figure 3: Visual comparison of the denoising task on the "Butterfly" example among the Baseline (ReLU + Position Encoding), `TRIDENT`, SIREN, MFN, and WIRE methods.

We use the Peak Signal-to-Noise Ratio (PSNR) as the metric to evaluate the experiment and train with 2K iterations with different nonlinearities. We use $l_2$ loss between the INR output and the input image. Figure 2 shows the denoising result of each nonlinearity.

Figures 2&3 show the denoising result for the parrot and butterfly images. Our method not only achieves the best performance in both cases but is also the only one that effectively preserves detailed features, demonstrating its robustness to noise. MFN and WIRE rank as the second-best, generally recovering the images but falling short in detail preservation. SIREN and the Baseline (ReLU+POS.ENC.) perform the least effectively, often resulting in over-smoothed outcomes. These examples underscore our method's robustness to noise, attributable to its spatial compactness, which is crucial for capturing important information concentrated in specific image regions.

### 4.3 3D Occupancy Reconstruction

To evaluate our method's representativeness in a higher dimension, we learn to represent the occupancy volume of a 3D image, the Thai statue. We sample the 3D image into a $512^3$ grid, assigning a value of 1 to pixels outside the 3D shape and 0 to those inside. For the marching cubes algorithm, we set the threshold at 0.5. The Intersection over Union (IoU) value was used as a metric to measure the reconstruction level. During the experiments, we employed the Mean Squared Error (MSE) as the loss function and trained the model with 4 different nonlinearities for 300 iterations.

Figure 4 visualises the results of the 3D Occupancy Reconstruction. `TRIDENT` achieves a 99.29% score, nearly perfect, successfully reconstructing all fine details of the 3D Thai statue input. WIRE and the baseline (ReLU+ POS. ENC.) attain scores between 98% and 99%, but lose some details during training. SIREN and MFN, with results ranging between 97% and 98%, demonstrate less representational efficacy as the dimensionality increases. This experiment highlights `TRIDENT`'s superior representativeness not only for low-order features but also for high-order features.

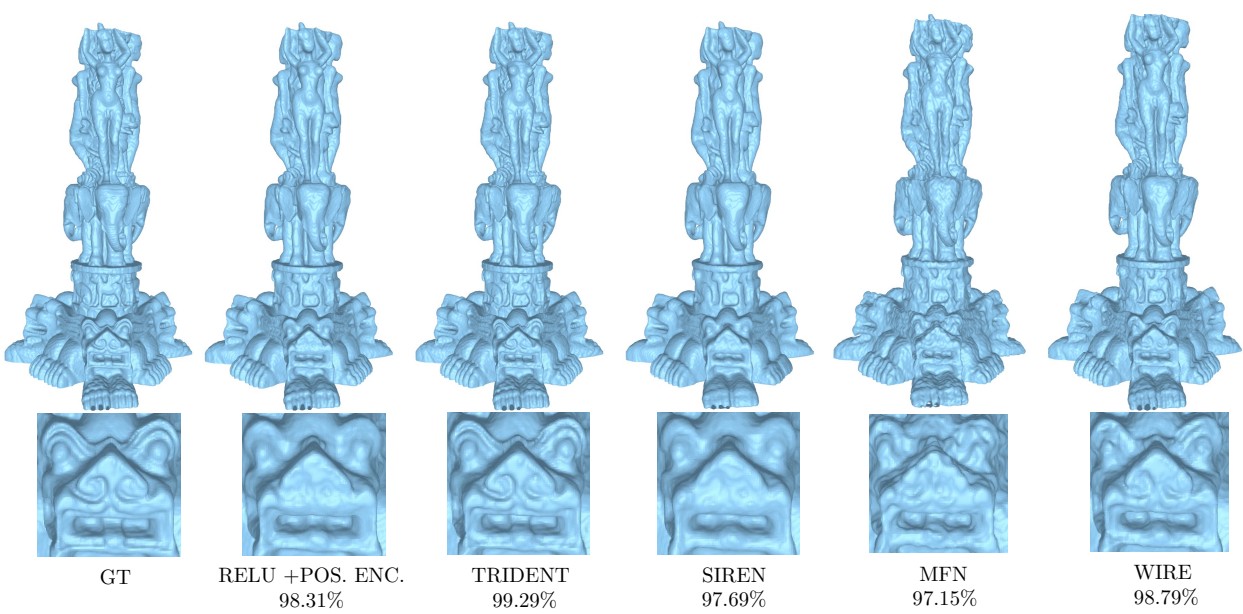

Figure 4: The visualisation comparison of 3D Occupancy Reconstruction (4×) task on "Thai-Statue" example among the Baseline (ReLU + Position Encoding), `TRIDENT`, SIREN, MFN, and WIRE methods.

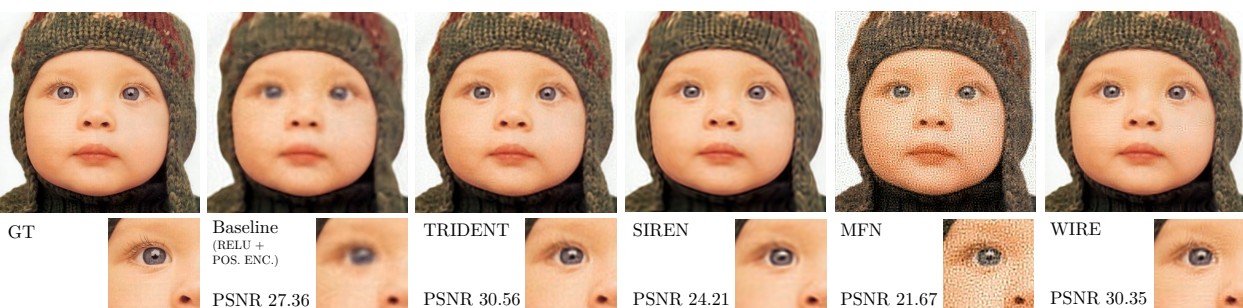

Figure 5: Visual comparison of single image super-resolution (4×) task on "Baby" example among the Baseline (ReLU + Position Encoding), `TRIDENT`, SIREN, MFN, and WIRE methods.

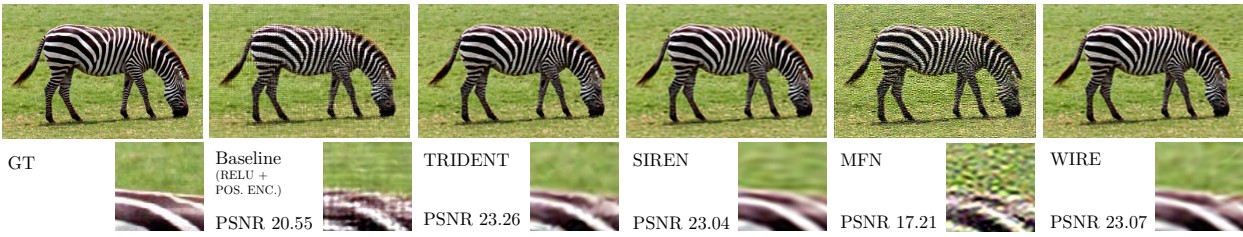

Figure 6: Visualisation comparison of single image super-resolution (4×) task on "Zebra" example among the Baseline (RELU + Position Encoding), `TRIDENT`, SIREN, MFN, and WIRE methods.

## 4.4 Single Image Super Resolution

In the task of image super-resolution, we aim to construct a high-resolution image from a low-resolution one. INR's high representativeness is particularly beneficial for super-resolution. We used the well-known datasets Set5 (Bevilacqua et al., 2012) and Set14 (Zeyde et al., 2012). We implemented 4× super-resolution

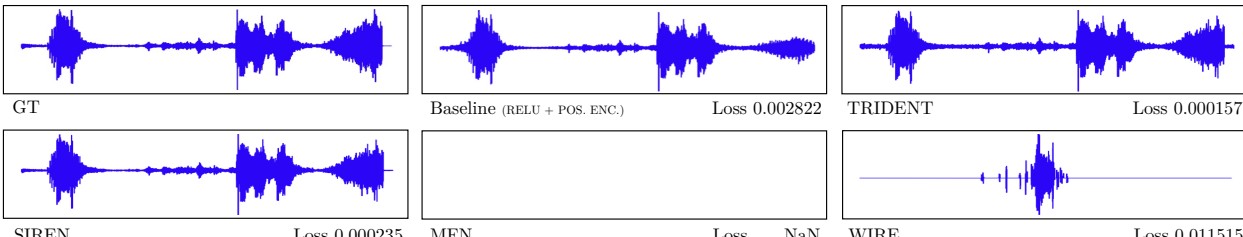

Figure 7: Visual comparison of audio reconstruction task on "Friends" example among the Baseline (ReLU + Position Encoding), TRIDENT, SIREN, MFN and WIRE methods.

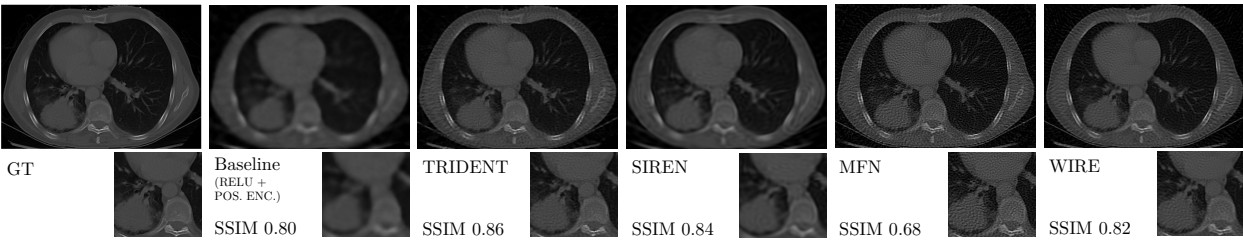

Figure 8: Visualisation comparison of CT reconstruction task on 100 projections among the Baseline (ReLU + Position Encoding), TRIDENT, SIREN, MFN and WIRE methods.

on four images randomly selected from these two datasets. For each task, we trained the model for 2000 iterations, using the $l_2$ loss between the input low-resolution picture and the INR output. We also employed the Peak Signal-to-Noise Ratio (PSNR) between the INR output and the ground truth to assess the quality of the images. Our method outperformed the other nonlinearities in all four tasks. Figure 5 presents the 4× super-resolution results for 'the baby' image using different nonlinearities, and Figure 5 visualises these results.

TRIDENT achieved the highest average PSNR (26.61 dB) and produced a smooth result with no obvious noise. WIRE and SIREN almost completely recovered the original images, albeit with some colour striation in the output. The baseline (ReLU+ POS. ENC.) was unable to fully recover the 'zebra' sample but performed relatively well with the 'baby' image. However, MFN showed the least effective performance in both images. Across these tasks, TRIDENT demonstrated its strong frequency compactness by effectively recovering both high-frequency features (as in the zebra task) and low-frequency features (as in the baby task) compared to the others, showcasing its stability and robustness during training to avoid colour striations.

### 4.5 Audio Reconstruction

Neural networks have shown the capability to represent audio signals (Agustsson & Timofte, 2017). However, it is hard for an MLP to solve the inverse problem of audio because it contains many kinds of nonlinearities including order and frequency features. To showcase the high representativeness and multimodality of our method compared to others, we aim to represent audio signals using our INR model. We evaluated the performance of MLPs with different nonlinearities on a selected four-second audio clip from the TV show 'Friends'. This clip features not only Joey's speech but also the background laughter of the audience. The model was trained using the Mean Squared Error (MSE) loss between the INR output and the original audio over 9000 iterations. We employ a 5-layer MLP with 256 hidden units in each layer for this task.

Figure 7 shows the visualisation of the audio signal we successfully recovered. Apart from SIREN and TRIDENT, other INR approaches failed to reconstruct the waveforms accurately. Pos+ReLU can represent the speech of Joey's speech but failed to capture the audience's laughter (particularly in the final $\frac{1}{4}$ of the waveform). WIRE resulted in an incorrect audio file, and MFN was unable to construct a waveform due to issues with gradient descent. Only our model and SIREN were able to accurately reconstruct the audio signal, with our

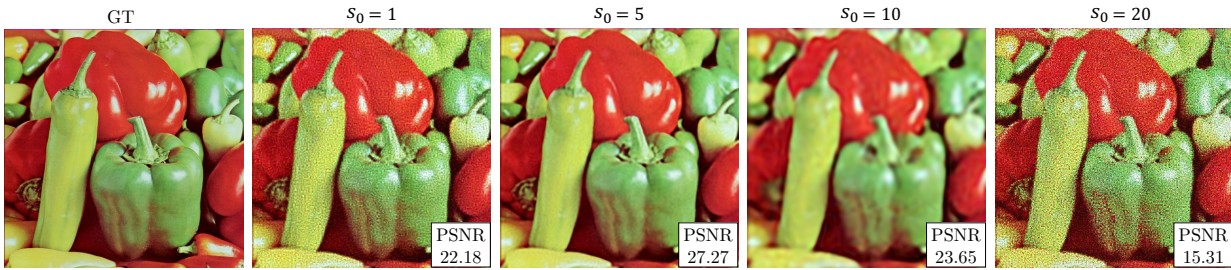

Figure 9: The 4× super-resolution result for `TRIDENT` with $s0 = 1, 5, 10, 20$ separately.

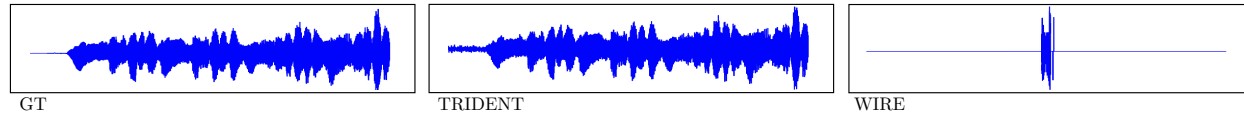

Figure 10: The visualisation of audio wave on a musical example "Bach's Cello Suite No.1" after reconstruction by `TRIDENT` and WIRE methods.

model achieving a lower loss than SIREN. This experiment highlights the exceptional representativeness of our method, benefiting from the *Nonlinear Trilogy*, where order compactness and frequency compactness aid in detailed recovery, and spatial compactness on specific regions of the signal.

### 4.6 CT reconstruction

To further test our method's efficacy in solving underconstrained problems with priors, we conducted an experiment focused on reconstructing a $256^2$ CT chest image (Clark et al., 2013), a challenging task for MLPs due to the presence of both even and odd features in the image. The CT reconstruction task utilises X-ray projections obtained from multiple angles to reconstruct internal structures through Radon transform, aiming for high-fidelity imaging with reduced data (Beister et al., 2012). In the main experiment, we reconstructed the image with 100 CT measurements. We increase the hidden features from 256 to 300 with 2 hidden layers and train with 5000 iterations. SSIM between the ground truth and the INR output served as the metric to measure the restoration quality.

Figure 8 displays the visualisations of the results from different nonlinearities. Our method produces the sharpest image with clearly defined features. SIREN and WIRE also yield clear outputs, but their results include some striation artifacts. The baseline approach (ReLU +POS.ENC.) leads to a blurred image. MFN performed the worst in this task, resulting in an output image with excessive noise. This demonstrates that our method, incorporating both odd and even elements in its formula, serves as an effective representation tool for solving underconstrained inverse problems.

### 4.7 Further Experiments: Music Audio reconstruction

To further show the advantage of `TRIDENT` brought by the nonlinear trilogy: order, frequency, and spatial compactness, we choose a hard task for MLP, music reconstruction. The first 7 seconds from Bach's Cello Suite No.1 is selected as input and the setting is the same as described in Section 4.5.
Figure 10 shows the visualisation of the reconstruction results of Bach's music via `TRIDENT` and WIRE. TRIDENT inherits almost all features from the ground truth, while WIRE fails to preserve any of them. Through this ablation experiment, we further show `TRIDENT`'s high representativeness by the nonlinear trilogy.

### 4.8 Ablation Study: Choice of Parameter $s_0$

The parameter $s_0$ controls the length of the window. A too-narrow window reduces representativeness by losing low-frequency information and increasing sensitivity to rapid signal changes. Conversely, a too-large

window causes `TRIDENT` to lose spatial compactness and high-frequency information. Consequently, we conducted a 4× super-resolution experiment with varying $s_0$ values of 1, 5, 10, 20, with the same setting as in section 4.4. The input 'pepper' is sourced from the well-known super-resolution dataset Set14 (Zeyde et al., 2012).

The visualisation results are shown in Figure 9. With $s_0$ = 20, the image quality is poor due to the excessively large window. For $s_0$ = 10, the result is oversmoothed, losing high-frequency features. However, with $s_0$ = 1, the window is too narrow, leading to increased noise sensitivity. Our conclusion is that setting $s_0$ around 5 is optimal, achieving a PSNR of 27.27 and preserving both high-frequency and low-frequency information.

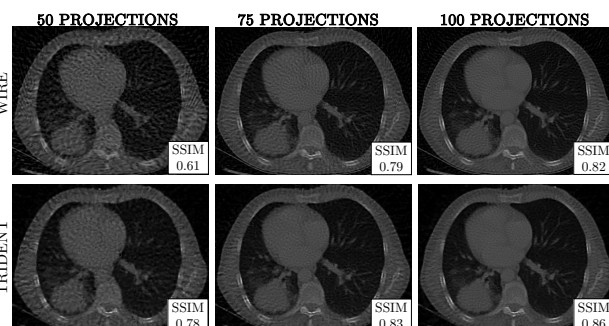

Figure 11: The CT reconstruction with 50, 75 and 100 projections.

### 4.9 Ablation: CT Reconstruction with Varying Number of Projections

To test the robustness of `TRIDENT` in solving inverse problems, we do an experiment on CT Reconstruction with a decreasing number of measurements with the same experiment setting and the same data of 4.6. The CT reconstruction results separately from 100 projections, 75 projections, and 50 projections are visualized in figure 11. In all projections, we hold a leading gap over WIRE, especially in lower projections (SSIM: 0.61 (WIRE) vs 0.78 (`TRIDENT`)). `TRIDENT` is more robust and the leading gap is larger with the dimension decreasing, which enables patients to reduce radiation exposure.

## 5 Conclusion

In this work, we have introduced `TRIDENT`, a new formula for Implicit Neural Representation, which benefits from the nonlinear trilogy: Order, Frequency, and Spatial Compactness. Strong theoretical principles are provided to show `TRIDENT`'s high representativeness and robustness. Various experiments are conducted to show `TRIDENT`'s versatility and high capacity for solving inverse problems, including CT, audio, 3D occupancy reconstruction, denoising, and super-resolution tasks. The performance of `TRIDENT`is underpinned by three foundational components: order compactness, frequency compactness, and spatial compactness. Together, these elements enable the model to capture variations across different orders of derivatives with high fidelity, ensuring an unparalleled ability to represent both broad features and intricate details. Moreover, `TRIDENT` demonstrates a remarkable capacity to accurately localise and render detailed information within specific spatial regions, showcasing its comprehensive approach to data representation. `TRIDENT` achieves the best among all the inverse problems that we tested on and hence gives the best solution for nonlinear INRs. While there remain unexplored inverse problems that push the boundaries of Implicit Neural Representations (INR), our solution stands at the forefront of the tasks already developed using INR. Exploring these uncharted areas represents a promising next step for the community.

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
