# OpenReview forum: "TRIDENT: The Nonlinear Trilogy for Implicit Neural Representations"
_TMLR — Rejected by TMLR_

### Review · Reviewer_o7eH · 2024-03-08

**Summary Of Contributions:**

The submission considers the problem of constructing implicit neural representations (INRs) instead of grid-based representations. INRs have been used in various domains in the past few years, such as image, audio, video processing, and robotics. The work introduces a new formula for implicit neural representations called TRIDENT. This expression combines three ideas: spatial, frequency, and order compactness. The submission provides a theoretical motivation for the formula, an explanation of how it can ensure the abovementioned types of compactness, and demonstrates experimentally that the introduced approach outperforms all other methods on various image and audio processing tasks.

**Audience:**

Yes

**Claims And Evidence:**

Yes

**Requested Changes:**

**Crucial**

C1. Add a more in-depth summary of the work and discussion of the method instead of the statement "TRIDENT achieves always the best and hence it is the best solution for nonlinear INRs." appearing in the conclusion. It would be particularly valuable to see a discussion of the method's potential limitations and possible further work.


C2. Clarifying the theoretical explanation

I had two questions left after reading through the theoretical explanation.

1. What is the derivation for the equation in the second line of equation (5)?

2. What exactly is meant by the "log-linear" strategy at the end of Section 3, and how is it used to get the final function?

I also have several smaller suggestions.

3. In section 3.1, I suggest explaining what $j$, $\mathbf{W}$, and $\mathbf{b}$ are and clearly stating which parameters are trained and which are chosen in advance.

4. As I understand, the first $x$ appearing in (1) should be in bold.

5. I suggest replacing "without losing generality" with "without loss of generality.", which is the usual expression.

**Minor (Not critical to securing my recommendation)**

C3. Improving the writing

1. Section 5 organization and naming

Section 5.1 discusses a hyperparameter search for the value of $s_0$, so it is confusing that it is a subsection in the section titled "Ablation study."

It is also unclear why the experiments in Sections 5.2 and 5.3 must be moved to a separate section from the experiments in Section 4 and why they are called ablation studies. Similar to the experiments in Section 4, they essentially compare the performance of TRIDENT to other methods, specifically to WIRE.

2. p. 1: "...across diverse problem domains[ref]" --> missing reference.

3. p. 2, Section 2: "the authors of that Tancik et al. (2020) proposed" --> "the authors of Tancik et al. (2020) proposed"

4. p. 2, Section 2: "Inspired by the principles of Fourier Transformation, the authors introduced ..." --> "the authors of (Sitzmann et al., 2020) introduced ..."

5. p.4, after equation (3): "... which we call it Order Compactness" --> "... which we call Order Compactness"

6. p.4, after equation (6): "... in a well-localised in the spatial domain." --> unclear phrase.

7. p. 5, end of Section 3: "Using WIRE (Saragadam et al., 2023), we demonstrated the loss of high-order information without order compactness." --> It is unclear how WIRE is used to demonstrate that.

8. p.5, Section 4.1: "we all added" --> unclear phrase.

9. p.7, end of page: Table 5 --> Figure 5.

10. Citation for the Adam optimizer: (Kinga et al., 2015) --> (Kingma et al., 2015)

11. I suggest adding somewhere that SSIM stands for Structural SIMilarity, maybe where the paper for SSIM is cited.

12. Figures 2 - 8 and 10 use OURS in the description, while in plots and the description of other figures, it is TRIDENT. I suggest changing OURS to TRIDENT everywhere to avoid confusion.

13. In general, I suggest one additional pass through the paper to polish the writing and fix typos.

**Strengths And Weaknesses:**

**Strengths**

S1. The experimental results are impressive. Experiments cover a wide range of applications in image and audio processing, and the suggested method noticeably outperforms other methods in all presented experiments.

S2. The paper is generally easy to follow and is mostly well-written, especially the experiment sections.

**Weaknesses**

W1. The suggested method is not discussed enough. The conclusion simply states that "TRIDENT achieves always the best, and hence it is the best solution for nonlinear INRs."

W2. (Minor) Although the general flow of the paper is easy to follow, the writing needs some polishing. After reading the theoretical explanation, I also had several small questions left.

---

> ### Author Response · Authors · 2024-03-28
> **Response to Reviewer o7eH**
>
> ➡️ **Add a more in-depth summary of the work and discussion of the method instead of the statement "TRIDENT achieves always the best and hence it is the best solution for nonlinear INRs." appearing in the conclusion. It would be particularly valuable to see a discussion of the method's potential limitations and possible further work.**
>
> We thank the reviewer for their valuable feedback. In response, we have updated our manuscript to better articulate the advantages of TRIDENT, emphasising its integration of order compactness, frequency compactness, and spatial compactness for superior performance in nonlinear INRs. Additionally, we've introduced a discussion on the method's limitations and potential directions for future research. The changes can be found in blue colour.
>
>
>
>
> ➡️ **Clarifying the theoretical explanation**
> * **What is the derivation for the equation in the second line of equation (5)?**
>
> We are happy to provide detailed derivation of
>
> \begin{equation}
> \begin{aligned}
> \sum_{n=0}^{\infty} \frac{(-1)^{n}}{2^{n} n !} \cos ^{n}(2 x)+\frac{(-1)^{n+1}}{2^{n+1} n !} \cos ^{n}(2 x)
> %+\frac{(-1)^{n+2}}{2^{n+2} n ! 2!} \cos ^{n}(2 x) +\frac{(-1)^{n+3}}{2^{n+3} n ! 3 !} \cos ^{n}(2 x)
> +\cdots
> = \sum_{i=0}^{\infty} \frac{(-1)^i}{2^{i} i !}
> \left[\cos^{i}(2x)+\binom{i}{1}\cos^{i-1}(2x)+\cdots\right]
> \end{aligned}
> \end{equation}
>
>
> \begin{array}{l}
> LHS
> =\sum_{n=0}^{\infty}\left(\sum_{j=0}^{\infty} \frac{(-1)^{n+j}}{2^{n+j} \cdot(n !) \cdot(j !)}\right) \cos ^{n}(2 x)
> \end{array}
> \begin{array}{l}
> =\sum_{n=0}^{\infty}\left(\sum_{j=0}^{\infty} \frac{(-1)^{n+j}}{2^{n+j}(n+j) !} \frac{(n+j) !}{n ! j !} \cos ^{n}(2 x)\right)
> \end{array}
>
> Let  $i=n+j$ , then
> \begin{array}{l}
>  =\sum_{i=0}^{\infty}\left( \sum_{j=0}^{i}\frac{{i!}}{(i-j)! j!} \cos ^{j}(2 x)\right) \cdot \frac{(-1)^{i}}{2^{i} i !}
> \end{array}
> \begin{array}{l}
> =\sum_{i=0}^{\infty}\left(\cos ^{i}(2 x)+
> \binom{i}{1}
> \cos ^{i-1}(2 x)+\cdots\right) \frac{(-1)^{i}}{2^{i} i !} \\
> =RHS
> \end{array}
>
>
>
>
> * **What exactly is meant by the "log-linear" strategy at the end of Section 3, and how is it used to get the final function?**
>
> The 'log-linear' strategy at the end of section 3 is introduced to adjust the frequency parameter $\sigma$ that we introduced to get the final function. We trained the model by adjusting the parameter $\sigma$ to maximise the likelihood of the training data with Adam optimisation algorithm.
>
>
> * **In section 3.1, I suggest explaining what j, W, and b are and clearly stating which parameters are trained and which are chosen in advance.**
>
> The parameter $j$ is the mapping index with range $1$ to $m$, while $W$ and $b$ are the parameters used in the neural network, as weight and bias. We thanks the reviewer to pointing out the confusion, and updated a clarification in blue in the updated manuascript.
>
>
> * **As I understand, the first x appearing in (1) should be in bold.**
>
> The $x$ in equation 1 should indeed be in bold as $\mathbf{x}$, we thank the reviewer for pointing out the typo. The modification has been made in blue in the updated document.
>
>
>
> * **I suggest replacing "without losing generality" with "without loss of generality.", which is the usual expression.**
>
> We thank the reviewer for the suggestion, it is updated accordingly in blue.
>
>
>
>
> ➡️ **Minor (Not critical to securing my recommendation)**
>
> Thanks for the insighful comments. We have updated the manuscript addressing every single minor suggestion. All changes can be found in blue colour.

---

> > ### Comment · Reviewer_o7eH · 2024-04-10
> > **Further comments**
> >
> > I thank the authors for their explanations and changes to the paper. I have checked the updated manuscript, and while the points I raised have been satisfactorily resolved and my questions answered, I have several minor comments on the updated version.
> >
> > 1. Continuing the question about the log-linear approach, it is confusing that the method used to determine the parameter mentioned essentially inside the sentence about the formula derivation. I suggest rephrasing the sentence before the last equation on page 5 or adding a sentence or two explaining it clearly in the paper.
> >
> > 2. I found the addition of Table 1 helpful. However, it now has some content that needs to be clarified. Specifically, why is the Wavelet transform an advantage and the ReLU function a disadvantage? I suggest listing the consequences of using these approaches instead. For instance, as I understand, instead of ReLU activation, the disadvantage of the FourierNet should be poor representation in the high-frequency domain. A more detailed explanation, for example, clarifying that it happens due to the ReLU activation, can be added to the table caption or in the text if needed.
> >
> > 3. In equation (5), the RHS of the first and second lines in black can be removed. The RHS of the first line is the same as the first line in blue, and the second line is not used.
> >
> > 4. The last sentence of the conclusion is hard to parse because of several typos: "While there are still unexplored inverse problems pushing the boundaries of INR, our solution remains at the forefront of the already developed tasks using INR, but would be promising next step to challenge on the other inverse problem that yet to be explore in the community."
> > I suggest rephrasing it to improve clarity. For example, something like "While there are still unexplored inverse problems pushing the boundaries of INR, our solution remains at the forefront of the already-developed tasks using INR. Nonetheless, it would be a promising next step to challenge the other inverse problems yet to be explored in the community.".

---

> > > ### Author Response · Authors · 2024-04-10
> > > **Responses_2**
> > >
> > > ➡️ **Continuing the question about the log-linear approach, it is confusing that the method used to determine the parameter mentioned essentially inside the sentence about the formula derivation. I suggest rephrasing the sentence before the last equation on page 5 or adding a sentence or two explaining it clearly in the paper.**
> > >
> > >
> > > We thank the reviewer for their suggestion. To address this comment, we have updated the manuscript to clarify the introduction of a frequency parameter $\sigma$  and the application of a log-linear strategy to enhance our representation's efficiency. This strategy is designed to balance magnitudes of different scales and aims to improve the output.
> > >
> > >
> > > ➡️ **I found the addition of Table 1 helpful. However, it now has some content that needs to be clarified. Specifically, why is the Wavelet transform an advantage and the ReLU function a disadvantage? I suggest listing the consequences of using these approaches instead. For instance, as I understand, instead of ReLU activation, the disadvantage of the FourierNet should be poor representation in the high-frequency domain. A more detailed explanation, for example, clarifying that it happens due to the ReLU activation, can be added to the table caption or in the text if needed.**
> > >
> > > We thank the reviewer for their insightful comment. We concur that the label "Advantage" in Table 1 may not adequately convey the intended message. Thus, we have revised this to "Principle" to more accurately reflect the core characteristics of each approach. Pertaining to the disadvantage associated with FourierNet, we have refined the description to "Frequency Limitation" to encapsulate the well-known challenges related to ReLU activation's handling of high-frequency components. All changes can be found in blue colour.
> > >
> > >
> > > ➡️ **In equation (5), the RHS of the first and second lines in black can be removed. The RHS of the first line is the same as the first line in blue, and the second line is not used.**
> > > We thank the reviewer for the suggestion. In the updated version, we have removed the suggeted lines.
> > >
> > > ➡️ **The last sentence of the conclusion is hard to parse because of several typos: "While there are still unexplored inverse problems pushing the boundaries of INR, our solution remains at the forefront of the already developed tasks using INR, but would be promising next step to challenge on the other inverse problem that yet to be explore in the community." I suggest rephrasing it to improve clarity. For example, something like "While there are still unexplored inverse problems pushing the boundaries of INR, our solution remains at the forefront of the already-developed tasks using INR. Nonetheless, it would be a promising next step to challenge the other inverse problems yet to be explored in the community.".**
> > >
> > > We thank the reviewer for their feedback. We agree that the readability of this part was challenging. We have made updates to clarify our message.

---

### Review · Reviewer_wgCG · 2024-03-14

**Summary Of Contributions:**

The paper proposes a novel implicit neural representation form, based on several principles that help to incorporate information about features of different order, frequency and spatial position. The effectiveness of the proposed INR is demonstrated on several tasks, including super resolution, audio reconstruction and spatial reconstruction and compared to several baselines.

**Audience:**

Yes

**Broader Impact Concerns:**

No concerns.

**Claims And Evidence:**

Yes

**Requested Changes:**

1 - Please add the explanation of the approach in general, frame it in the state of the art reconstruction approaches. Maybe even evaluation compared to them would be interesting (if applicable).

2 - I would argue that showing image of the sound wave is not efficient for understanding the results (fig.7 and fig.10). Maybe just providing link to the audiofiles is better.

3 - Ablation study shows that the approach is very sensitive to some hyperparameters, but no hints on how to adjust them are not given.

4 - Please clarify what are j and m stand for in formula 1 and also what should be there in the ... parts.

Minor:

1 - Typo in the reference to Kingma et al. in section 4.1

2 - In equation 5 the summation index switches from n to i

3 - It seems that there is a typo in the last sentence of section 3.1, since \phi_i is not taking part in definition of \gamma

4 - Missing reference in the third paragraph of the introduction ([ref])

**Strengths And Weaknesses:**

Proposed approach is explained with mathematical justifications and empirically evaluated on several tasks.

Due to the novelty of the field of INRs it would be very useful to add more explanations in the introduction about the purpose of such representations, how exactly neural networks are related to it, how this field is related to state-of-the-art generative models that can also perform reconstruction tasks. Without these explanations the methodology is very hard to understand.

---

> ### Author Response · Authors · 2024-03-28
> **Response to Reviewer wgCG**
>
> ➡️ **Please add the explanation of the approach in general, frame it in the state of the art reconstruction approaches. Maybe even evaluation compared to them would be interesting (if applicable).**
>
> Thank you for your valuable feedback and suggestion to elaborate on our approach within the context of state-of-the-art reconstruction methodologies. We agree that a comprehensive explanation and framing within the current landscape would enhance the manuscript and appreciate the opportunity to clarify how our approach distinguishes itself.
>
> In our work, we introduce a new approach that is characterised by a unique mathematical components: frequency compactness, spatial compactness, and order compactness. To the best of our knowledge, this triad has not been collectively considered in the context of implicit neural representations. Frequency compactness allows our model to effectively represent and reconstruct signals at various scales, improving the fidelity of high-frequency details. Spatial compactness ensures that our representations are localized and precise, facilitating the accurate rendering of intricate spatial variations. Lastly, order compactness contributes to the smoothness and continuity of the represented data, which is particularly beneficial for complex geometries and textures.
>
> In our current work, we compared our TRIDENT technique against the state-of-the-art techniques. However, to provide more clarity, we have done the following changes.  Firstly, we added a clarifying notes on the conceptual benefit of our trilogy in Section 3 (see changes in blue colour). Secondly, we have added a new table displaying the advantages and disadvatanges of existing INRs techniques and ours.
>
>
>
> ➡️ **I would argue that showing image of the sound wave is not efficient for understanding the results (fig.7 and fig.10). Maybe just providing link to the audiofiles is better.**
>
> We thank the reviewer to point a great option for efficiently interpreting the results, we have attached the folder links here with all the audio files.
>
> For the audio reconstruction in Fig.7:
> https://drive.google.com/drive/folders/1i7t3_z9FhJu8hbBZFQOJJESWBgj9LxgT?usp=drive_link.
>
> For the audio reconstruction in Fig.10:
> https://drive.google.com/drive/folders/113YPlovkwA14WGu3o-_K8rOJe-KEErjz?usp=drive_link.
>
> We will include the audio results directly in our project website, which will be available upon acceptance due to anonymous reason.
>
>
>
> ➡️ **Ablation study shows that the approach is very sensitive to some hyperparameters, but no hints on how to adjust them are not given.**
>
> Thanks for the comment. We adhere to a standard protocol by employing a grid search technique to identify the optimal hyperparameter values for our approach. This methodical search allows us to systematically explore a range of possible values and evaluate their impact on the performance of our model. By doing so, we ensure that the chosen hyperparameters are well-suited to maximise the efficacy of our representation method. We therefore provide a set of recommended hyperparameter values that we found to yield the best results in our experiments. To address this comment, we have added a clarifying note in Subsection 4.1 in blue colour.
>
>
>
> ➡️ **Please clarify what are j and m stand for in formula 1 and also what should be there in the ... parts.**
>
> Thanks for the comment. We clarify that $m$ stands for the mapping size and $j$ stands for the mapping index, ranging from $1$ to $m$, while the dots in equation 1 represent the terms with different choices of $j$. We clarify all the notations in the updated manuscript in blue.
>
>
>
> ➡️ **Typo in the reference to Kingma et al. in section 4.1**
>
> We thank the reviewer for pointing out the typo, the change has been made in our updated manuscript in blue.
>
>
>
> ➡️ **In equation 5 the summation index switches from n to i**
>
> The summation index was changed due to a change of variable during derivation. To clarify the process, we add additional derivation steps in equation 5 in blue in the updated manuscript.
>
>
>
> ➡️ **It seems that there is a typo in the last sentence of section 3.1, since \phi_i is not taking part in definition of \gamma**
>
> We thanks the reviewer for pointing out the typo, it should be ‘and’ not ‘where’, the correction has been made in blue in the updated version.
>
>
>
> ➡️ **Missing reference in the third paragraph of the introduction ([ref])**
>
> We thanks the reviewer for pointing out the typo, the additional reference had been made in blue in the updated version.

---

> > ### Comment · Reviewer_wgCG · 2024-04-04
> >
> > Thank you for the answers.
> >
> > I would clarify again, that by framing the approach in the current state-of-the-art I rather meant approaches different from INR. I understand that it is a novel field, so comparison within the field will not help to grasp the difference between previously used representations from deep networks.
> >
> > Moreover, the clarity of the algorithm can be also helpful - how exactly DNN is employed in such approaches.

---

> > > ### Author Response · Authors · 2024-04-10
> > > **Responses_2**
> > >
> > > ➡️ **I would clarify again, that by framing the approach in the current state-of-the-art I rather meant approaches different from INR. I understand that it is a novel field, so comparison within the field will not help to grasp the difference between previously used representations from deep networks.
> > > Moreover, the clarity of the algorithm can be also helpful - how exactly DNN is employed in such approaches.**
> > >
> > > Thank you for your question. We appreciate the opportunity to clarify. Implicit Neural Representations (INRs) inherently differ from traditional Deep Neural Networks (DNNs) by focusing on learning from a single data instance, rather than relying on extensive training datasets. This distinction underpins the unique capability of INRs to model continuous, resolution-independent representations directly from individual instances. Consequently, comparing INRs directly with DNNs, which are designed and optimised for learning from large datasets, may not provide a fair or meaningful assessment of their respective capabilities and advantages. Our approach aligns with the protocol in the INR domain, emphasising the specialised application of INRs and the different philosophical foundation compared to traditional DNNs. As INRs are solely based on MLPs (also known as coordinate MLPs). We clarify this distinction in the related work in blue colour.

---

### Review · Reviewer_hF2P · 2024-03-15

**Summary Of Contributions:**

I apologize for being a non-expert in this topic of implicit neural representations, but I honestly don't think I understood the contributions of this paper. Perhaps the best course of action is for me to ask a list of questions below to first clarify all of my initial confusions, and possibly have a follow up discussion, before I finally update my review with strengths and weaknesses, as well as recommended changes.

### Questions

1. How is an implicit neural representation defined, and what is the purpose? I hope the authors can provide some additional context for readers who are not experts. At the same time, what is a traditional grid based approach?

2. Can you help me define the notations used in section 3? For example, what is the variable $x$, and what are the dimensions, what do the dots in equation 1 represent, and what is the output dimension of this function? What is the difference between $x$ and bold $\mathbf{x}$?

3. You introduced several concepts such as "order compactness" and "frequency compactness", but how are these terms defined precisely? It's unclear for me what these terms mean.

4. Towards the end of section 3, you demonstrated an example of using INR in Figure 1. I'm not sure what you did here, as in what were the input and output of this experiment?

Perhaps this is good enough to start for the first iteration of discussions. One small thing: I do want to briefly discuss the section headings with the finger pointing emoji. I'm not sure what the TMLR formatting requirements are, but I found these emoji distracting and unprofessional. Regardless of the formatting requirements, I would recommend you avoid using these, so the readers can focus on the content.

**Audience:**

Yes

**Claims And Evidence:**

No

**Requested Changes:**

N/A

**Strengths And Weaknesses:**

N/A

---

> ### Author Response · Authors · 2024-03-28
> **Response to Reviewer hF2P**
>
> ➡️ **How is an implicit neural representation defined, and what is the purpose? I hope the authors can provide some additional context for readers who are not experts. At the same time, what is a traditional grid based approach?**
>
> We thank the reviewer for the comments. We would like to start by clarifying the concept of Implicit Neural Representations (INR), which serve as a modern method for encoding data structures or continuous signals using neural networks. Unlike traditional grid-based approaches that store information in discrete intervals, INRs represent data in a continuous, memory-efficient form, enabling more control and manipulation of the represented content.
> Secondly, we now clarify that a traditional grid-based approach, on the other hand, represents data on a fixed, discrete grid, like pixels for images or voxels for 3D shapes. This can be less efficient and more memory-intensive, especially for high-resolution data or complex geometries. That is why, INR can be used a wide range of applications. To address this comment, we have updated the text, and the changes can be found in the first paragraph of Section 1 in blue colour.
>
>
> ➡️ **Can you help me define the notations used in section 3? For example, what is the variable x, and what are the dimensions, what do the dots in equation 1 represent, and what is the output dimension of this function? What is the difference between x and bold x?**
>
> Thanks for the comment. We clarify that the variable $\mathbf{x}$ defines the input signal to each neuron in the MLP layer, and the dimension of $\mathbf{x}$ is the dimension of the input data according to different tasks. The dots in equation 1 represent the terms with different choices of $j \in$ {$1,2,...,m$}. The $x$ in the later part of the methodology denotes general input into a function, while the bold $\mathbf{x}$ in equation 1 and 2 denotes the data with the same dimension as input data. The $x$ in equation 1 should be the bold $\mathbf{x}$, we thank the reviewer for pointing out the typo. We have added a clarifying note to make this passage clearer.
>
>
> ➡️ **You introduced several concepts such as "order compactness" and "frequency compactness", but how are these terms defined precisely? It's unclear for me what these terms mean.**
>
> We thank to the reviewer for the comment. To addres this comment, we provide detailed context on these terms.
>
> Firstly, we refer to order compactness to the ability of a representation to encapsulate information efficiently across different orders of derivative or variation within the data. Moreover, 'order' typically relates to the smoothness and the level of detail that the representation can capture. . Mathematically, this can be associated with the representation's capacity to approximate functions and their derivatives to a certain degree of accuracy.
>
> Secondly, frequeny compatness refer to  describe the efficiency with which a representation can encapsulate information at different frequencies.  A representation with high frequency compactness can accurately reproduce both the overarching structures and the intricate details of the data, making it particularly valuable for applications requiring high fidelity.
>
> To add clarity, we have updated the manuscript to include an explicit discussion on these concepts.  The changes can be seen in Section 3 in blue colour.
>
>
> ➡️ **Towards the end of section 3, you demonstrated an example of using INR in Figure 1. I'm not sure what you did here, as in what were the input and output of this experiment?**
>
> Thanks for the comment, we provide next details on the setting for Figure 1. The purpose of this figure is to illustrate the effect of our trilogy. The experiment is on CT reconstruction task using 100 projections to support our nonlinear trilogy: TRIDENT (trilogy), and TRIDENT without Frequency, Order, and Spatial Compactness separately.
> To address this comment, we have done the following changes. Firstly, we have added clarity in the setting used for the CT experiments. Secondly, we have reworded the description on the last paragraph of subsection 3.2 for clarity purposes. All changes can be seen in blue colour.
>
>
> ➡️ **One small thing: I do want to briefly discuss the section headings with the finger pointing emoji. I'm not sure what the TMLR formatting requirements are, but I found these emoji distracting and unprofessional. Regardless of the formatting requirements, I would recommend you avoid using these, so the readers can focus on the content.**
>
> We acknowledge that style preferences can vary significantly among readers and reviewers. It's worth noting that other reviewers have not commented on this aspect, which might suggest a range of opinions on such stylistic choices. However, we deeply respect the scholarly standards and the reading experience of all reviewers and readers.
>
> If you have strong feelings on this matter, we are more than willing to reconsider our stylistic choices to ensure that our manuscript.

---

> > ### Comment · Reviewer_hF2P · 2024-04-04
> > **Further Questions**
> >
> > Thank you for the detailed response. I have quite a few follow up questions regarding this manuscript that I hope you can still address.
> >
> > 1. I still don't fully understand what you are referring to as an implicit neural representation. Is the point that you want to represent a dataset using a continuous function that interpolates the dataset? What does the output here mean? Can you give an example of what this representation is doing with respect to a dataset? What would the grid-based approach do for this example?
> >
> > 2. Is $\Phi(x)$ what you are defining as the implicit neural representation? How are the parameters $\sigma, W, b$ chosen for your applications?
> >
> > 3. What's the significance of order compactness and frequency compactness? My point here being that yes this is a property, but why do you need this property and why would it help? In particular, I don't think the Stone--Weierstrass Theorem is a convincing reason, since Fourier series is can also approximate arbitrary functions in a compact interval.
> >
> > 4. For Figure 1, I was hoping to get a full description of the experiment setup. E.g. what is the dataset here? What do you mean by the first 100 projections and how did you compute it? How did you fit the model? What exactly is the output and what does it mean? This is important because even a non-expert should be able to replicate your experiment from reading the paper.

---

> > > ### Author Response · Authors · 2024-04-10
> > > **Responses_2**
> > >
> > > ➡️ **I still don't fully understand what you are referring to as an implicit neural representation. Is the point that you want to represent a dataset using a continuous function that interpolates the dataset? What does the output here mean? Can you give an example of what this representation is doing with respect to a dataset? What would the grid-based approach do for this example?**
> > >
> > >
> > > Thanks for the question. We appreciate the opportunity to further clarify the concept and application of Implicit Neural Representations (INR) in our work. It's important to note that the concept of INR is not novel to our study but is a well-established area of research within the literature. Our goal is to contribute with a new solution within the widely studied area of INR.  We would like to address your questions separatly next, with updates in related work in blue colour.
> > >
> > > **[What we refer to INR?]** Firstly, at its core, an INR employs a neural network to model a continuous function that can interpolate or approximate the underlying data distribution of a dataset. This continuous function is learned from the discrete samples of the dataset, enabling the representation of complex data structures or spaces in a compact and continuous manner.
> > >
> > > **[Meaning of the output]** Secondly, the output of an INR typically represents the value of the continuous function at a given point in the input space. This can vary based on the application, such as generating the colour value at a specific pixel for image synthesis or the density at a point in space for 3D volume rendering.
> > >
> > > **[Example INR]** Thirdly, imagine we want to use an INR to model a 2D image. The network would learn to predict the colour of any given pixel (x, y)  on data. For a given input coordinate, the network outputs the colour value, effectively allowing us to reconstruct the image or even generate new parts of the image by querying points not explicitly represented in the original dataset.
> > >
> > > **[Grid-based example]** Unlike grid-based methods that discretely represent data and are limited by resolution and dimensionality, INRs provide a continuous, resolution-independent representation. For the same 2D image example, a grid-based approach might store colour values at fixed pixel locations, leading to potential issues with scaling or transformation. INRs, however, can theoretically generate any pixel value within the image domain, offering a more flexible and scalable solution.
> > >
> > >
> > >
> > > ➡️ **Is $\Phi(x)$ what you are defining as the implicit neural representation? How are the parameters $\sigma, W, b$ chosen for your applications?**
> > >
> > > Yes, $\Phi(x)$ defines as the function modelled by the neural network to represent continuously across a domain. The parameters $\sigma, W, b$ —which  represent the activation function, weights, and biases of the network, respectively—are crucial for defining the function's shape and complexity.  Training the network involves adjusting $W , b$   to minimise the difference between the network's output and the actual data. We have added a clarified note on this.
> > >
> > >
> > > ➡️ **What's the significance of order compactness and frequency compactness? My point here being that yes this is a property, but why do you need this property and why would it help? In particular, I don't think the Stone--Weierstrass Theorem is a convincing reason, since Fourier series is can also approximate arbitrary functions in a compact interval.**
> > >
> > >
> > > Thanks for the question. We now clarify that order compactness and frequency compactness relate to how efficiently a model can approximate complex functions within a specific domain. While Fourier series can indeed approximate arbitrary functions, the compactness properties enhance the efficiency and effectiveness of learning by reducing the model's complexity and improving generalisation. Essentially, these properties guide the selection of functions that the model can approximate well, focusing on those that are significant for the application. This focus helps in achieving higher fidelity, making the learning process more efficient and leading to better performance. We have added a clarified note in blue colour.

---

> > > > ### Author Response · Authors · 2024-04-10
> > > > **Resposes_2b**
> > > >
> > > > ➡️ **For Figure 1, I was hoping to get a full description of the experiment setup. E.g. what is the dataset here? What do you mean by the first 100 projections and how did you compute it? How did you fit the model? What exactly is the output and what does it mean? This is important because even a non-expert should be able to replicate your experiment from reading the paper.**
> > > >
> > > > We thank the reviewer for the comment. We would like to clarify that in the context of CT reconstruction, the term "projections" refers to the images obtained by passing X-rays through a patient from various angles.  These projections are then used to reconstruct the patient's internal structures. CT projections are computed using a process called Radon transform. The data used is publicly available where the full acquisition is provided and we chose a subset of 100 projections following the standard protocol for this dataset. The input is a set of projects, as the main goal is to recontruct a high fidelity with less data, and the output is the reconstructed image. For sake of clarity, we pointed out the full description, of the setting of this visualisation, in the experimental results and add a reference about the basics of CT reconstruction in blue colour.

---

> > > > > ### Comment · Reviewer_hF2P · 2024-04-22
> > > > > **Further Questions 2**
> > > > >
> > > > > Thank you for the responses, and apologize for the late reply on my part. I think I'm starting to understand what's happening. Just to clarify a bit further and I think I will be ready soon to make a recommendation.
> > > > >
> > > > > 1. So let's consider an image INR on a unit square $\Phi : [0,1] \times [0,1] \to \mathbb{R}^3$. If I understand correctly, using one single parameterization of the $W,b,\sigma$ (btw I think $\sigma$ the frequency right?), this function $\Phi$ can only represent one image right? So when you train the weights, do you only train one set of weights on a single image?
> > > > >
> > > > > 2. Related to the previous question, yes I now understand a bit better where the data comes from. So what is the pseudo algorithm for training? I might be missing something, but I genuinely do not see how I should be implementing the algorithm you are proposing in this paper.

---

> > > > > > ### Author Response · Authors · 2024-05-09
> > > > > > **Responses_3**
> > > > > >
> > > > > > ➡️ **So let's consider an image INR on a unit square $\Phi : [0,1] \times [0,1] \to \mathbb{R}^3$. If I understand correctly, using one single parameterization of the $W,b,\sigma$ (btw I think $\sigma$ the frequency right?), this function $\Phi$ can only represent one image right? So when you train the weights, do you only train one set of weights on a single image?**
> > > > > >
> > > > > > We thank to the reviewer for the comment. You are correct that the function $\Phi: [0,1] \times [0,1] \to \mathbb{R}^3$ represents a single image, with each pixel color (RGB) encoded by the function's output. When using a parameterisation of $W$, $b$, and $\sigma$ (where $\sigma$  represents frequency), this set of parameters will only encode one image.
> > > > > >
> > > > > > Training is conducted such that a single set of weights and associated parameters is optimised per image. Thus, each set is specific to the particular image on which it was trained.
> > > > > >
> > > > > >
> > > > > >
> > > > > > ➡️ **Related to the previous question, yes I now understand a bit better where the data comes from. So what is the pseudo algorithm for training? I might be missing something, but I genuinely do not see how I should be implementing the algorithm you are proposing in this paper.**
> > > > > >
> > > > > > We thank to the reviewer for the comment. We provide the pseudocode next.
> > > > > >
> > > > > >
> > > > > > ____________________________________________________________
> > > > > > Pseudocode
> > > > > > __________________________________________________________
> > > > > >
> > > > > > Input: $\mathbf{x}$, the coordinate of the input image $y$
> > > > > >
> > > > > > Output: Trained model $\Phi$ that can map coordinates $\mathbf{x}$ to pixel values $y$.
> > > > > >
> > > > > > Initialise network parameters $\mathbf{W}, \mathbf{b}, \sigma$
> > > > > >
> > > > > >  $\textbf{for}$ $i$ from $0$ to epoch_number $\textbf{do}$:
> > > > > >
> > > > > >  &nbsp;   &nbsp;   &nbsp;  #calculate the loss for the example denoising task
> > > > > >
> > > > > >  &nbsp;  $\qquad L(\Phi(\mathbf{x}), y)= \int_\Omega |\Phi(\mathbf{x})- y|^2 = \frac{1}{N}\sum_{j=1}^N|\Phi(\mathbf{x}_j)- y_j|^2$
> > > > > >
> > > > > >  &nbsp;   &nbsp;   &nbsp;  Update the gradients to the network parameters
> > > > > >
> > > > > > $\textbf{end}$
> > > > > > ____________________________________________________________

---

> > > > > > > ### Comment · Reviewer_hF2P · 2024-05-13
> > > > > > > **Response**
> > > > > > >
> > > > > > > Apologies for the slow reply. I did not have access to internet in the past week.
> > > > > > >
> > > > > > > I think I'm happy with the questions for now.
> > > > > > >
> > > > > > > I will however, request the authors to include a more precise description of the functions I asked about, e.g. make it clear that $\Phi$ is a map representing a single image only, with the inputs and outputs specified. Ideally, I would also like to see short description of how the training is done, so it would be replicable even for a non-expert.

---

### Comment · Action_Editor_oUSG · 2024-05-14
**questions**

The results in the paper seem very promising! I have several questions regarding the paper and hope the authors can help clarify them.

1.  **Terminology:** I am a bit confused by the new terms: order compactness, frequency compactness, and spatial compactness. In Fourier analysis, order and frequency usually refer to the same thing (higher order is equivalent to higher frequency), as $\cos^n(x) = a \cos(nx) + ...$. Therefore, I don't see the difference between "order compactness" and "frequency compactness."

    Regarding spatial compactness, the paper states: "Thanks to the Gaussian window, we can represent the signal or image in a well-localized spatial domain." Does "Gaussian window" refer to the hyperparameter  $s_0$ in equation 1, which essentially controls the coefficients of $\cos^n(x)$ or $\cos(nx)$ in the Taylor expansion? If so, I believe the new terminology might make things more complicated. Overall, I recommend not introducing new terminology unless absolutely necessary.

2.  **Relation to FourierNet:** This framework seems very relevant to FourierNet and the paper "Fourier features let networks learn high frequency functions in low dimensional domains." To my understanding, the FourierNet papers use random cosine and sine functions to map coordinates to high dimensions, and then apply ReLU MLP. In this paper, you consider a geometrically spaced (non-random) mapping coupled with cosine and sine functions, and then apply MLP with activation functions $\exp(-x^2)$. Is this correct?

    Could you clarify the relationship and differences between these approaches in the paper? Since FourierNet is highly related, I would appreciate it if you could add FourierNet experiments as another baseline (if possible). Additionally, the FourierNet paper specifically mentions "learning high frequency functions in low dimensional domains," which is also a key novelty advocated in this paper. Could you elaborate on what you mean by "frequency limitation" in the table?


Minor comments:

In the first paragraph of the introduction, please add a reference for "Implicit Neural Representations" [citation!].
In Table 1, please add references for each network in the first column. Additionally, could you clarify what you mean by "principle" in the second column? Combining "principle" and "disadvantage" seems a bit unusual.
The numbers reported in Figure 3 appear inconsistent with those reported in Figure 7 of the WIRE paper ("Wavelet Implicit Neural Representations"). Please clarify this discrepancy.

Thanks!

---

### Decision · Action_Editor_oUSG · 2024-05-31

**Recommendation:** Reject

**Comment:**

The paper received mixed reviews (leaning reject, leaning accept, and accept). One reviewer expressed strong concerns about the paper's readability, while another requested more explanation of the underlying principles and how this work differs from other neural representations. The third author reviewer noted the proposed formula from the paper is "well theoretically motivated" and checked the correctness of the deviation of it, and endorsed acceptance.

I reviewed the paper and raised several questions (regarding new terminology and differentiation from FourierNet), hoping the authors would clarify. Unfortunately, I have not received a response after two weeks.

Additionally, the paper makes strong claims about its method (e.g., "TRIDENT gives the best solution for nonlinear INRs") that require deep domain expertise to verify. Unfortunately, none of us possess such expertise to endorse these claims.

Overall, the paper presents promising results, but the claims are too strong and require deep expertise to validate. I recommend the authors soften their claims, address the concerns raised by reviewers and the AC, and resubmit the paper.

**Audience:**

Yes. Implicit Neural Representations (INRs) are a significant and increasingly popular research area within machine learning, with applications across a wide range of domains.

**Claims And Evidence:**

The paper introduces TRIDENT, a new function for implicit neural representation. TRIDENT consists of a trigonometric embedding, followed by a Gaussian function, and then an MLP with Gaussian activation functions. The authors argue for several theoretical advantages of this proposal, including Order Compactness, Frequency Compactness, and Spatial Compactness. Through extensive experiments across a wide range of domains, they demonstrate impressive and promising performance, concluding that "TRIDENT achieves the best results among all the inverse problems tested and hence gives the best solution for nonlinear INRs." See "Comment section" for evidence and more details.